# Immune Organs and Immune Cells on a Chip: An Overview of Biomedical Applications

**DOI:** 10.3390/mi11090849

**Published:** 2020-09-12

**Authors:** Margaretha A. J. Morsink, Niels G. A. Willemen, Jeroen Leijten, Ruchi Bansal, Su Ryon Shin

**Affiliations:** 1Division of Engineering in Medicine, Department of Medicine, Harvard Medical School, Brigham and Women’s Hospital, Cambridge, MA 02139, USA; m.a.j.morsink@student.utwente.nl (M.A.J.M.); n.g.a.willemen@student.utwente.nl (N.G.A.W.); 2Department of Developmental BioEngineering, Faculty of Science and Technology, Technical Medical Centre, University of Twente, Drienerlolaan 5, 7522 NB Enschede, The Netherlands; j.c.h.leijten@utwente.nl; 3Translational Liver Research, Department of Medical Cell BioPhysics, Technical Medical Centre, Faculty of Science and Technology, University of Twente Drienerlolaan 5, 7522 NB Enschede, The Netherlands; r.bansal@utwente.nl

**Keywords:** immune system, microfluidics, immune cells, organ-on-a-chip, 3D in vitro model, immune system-on-a-chip

## Abstract

Understanding the immune system is of great importance for the development of drugs and the design of medical implants. Traditionally, two-dimensional static cultures have been used to investigate the immune system in vitro, while animal models have been used to study the immune system’s function and behavior in vivo. However, these conventional models do not fully emulate the complexity of the human immune system or the human in vivo microenvironment. Consequently, many promising preclinical findings have not been reproduced in human clinical trials. Organ-on-a-chip platforms can provide a solution to bridge this gap by offering human micro-(patho)physiological systems in which the immune system can be studied. This review provides an overview of the existing immune-organs-on-a-chip platforms, with a special emphasis on interorgan communication. In addition, future challenges to develop a comprehensive immune system-on-chip model are discussed.

## 1. Introduction

The first mention of immunology dates back to Ancient Greece [1]. Thucydides described how there was no recurrence of the plague in those who already suffered from the disease in 430BC [2]. Hippocrates—commonly known as the father of medicine—portrayed nature as the primary doctor. Immune responses, such as a fever, play a vital role in fighting against diseases [3]. Over the past centuries, many scientists and philosophers wondered about the immune system. However, it was not until the late 1800s that immunology appeared as the ’science of the host defense’ [4]. Pasteur—who is considered the father of immunology—was the first to describe that bacteria could cause an infectious disease, which would be later known as germ theory. Afterwards, Erlich and Metchnikov disputed the existence of a humoral system of antibodies versus the existence of phagocytes in the immune system. Ehrlich described the role, formation and fundamentals of antibodies in immunology, whereas Metchnikov postulated the importance of phagocytes and the concept of “self and non-self”, referring to what belongs to the body and what does not. In 1908, they shared the Nobel Prize for their respective contributions to the field of immunology [4,5]. The research on antibodies progressed in the 1930s and 1940s. Research was focused on the formation and function of antibodies, as well as understanding the importance of the antigen–antibody complex [6,7]. In 1940, Burnet described acquired immunity for the first time. He showed that antibodies that are formed by immune cells upon recognition of foreign antigens would replicate within the cell and corresponding daughter cells, resulting in a heightened secondary immune response [8]. In the 1950s, Burnet postulated the clonal selection theory, which explains the multiplication of circulating lymphocytes in response to specific antigens [9]. The term immune system was first coined in the 1960s by combining the lymphatic system and immunity [10]. The history of research on the immune system is summarized in Figure 1A.

Research on the immune system has mostly been conducted in two-dimensional (2D) in vitro cell cultures or in vivo animal studies [11,12]. In vivo research is mostly conducted on small rodents, such as mice, before testing any drugs or therapies for humans [13]. Additionally, zebrafish models have been used extensively to model the innate immune response, as well as inflammatory and infectious diseases [14]. However, there are still significant differences between the immune systems of rodents and fish models with humans, which can have detrimental results for clinical trials [13,15]. In vitro research on 2D cell cultures is unable to fully emulate the physiological multicellular microenvironment due to its simplistic nature, and thus cannot fully recapitulate the in vivo situation [16]. Therefore, research on general 2D cell cultures started shifting towards three-dimensional (3D) systems since the 1980s [17]. Human cell-based 3D systems bridge the gap between standard 2D cultures and animal models and human clinical trials by mimicking a physiologically relevant microenvironment without ethical concerns regarding in vivo animal research [18]. Due to its relevance for major diseases and the development of targeted therapeutic strategies using antibodies and the immune system, the evaluation of the immune system using 3D culture systems such as organ-on-a-chip (OoC) models has steadily gained momentum, as can be seen in Figure 1B [19,20]. Research commenced in the early 2000s on small immunological components, with projects such as emulation of lymphatic valves on a chip [21], microfluidic-based immune cell separation within blood samples [22,23], recapitulation of T-lymphocyte migration on a microfluidic-based multichannel platform [24], and, lastly, describing neutrophil chemotaxis in a microfluidic gradient chamber [25]. Gradually the platforms increased in complexity, thereby improving the emulation of the immune system, resulting in the OoCs discussed in this review.

This review highlights the functions of the immune system and how they can be recapitulated in 3D culture systems. Moreover, we discuss state-of-the-art immune system-on-a-chip models that are currently available, detail the recent trend of incorporating immune cells in distinct OoC systems and reflect on what should be done in the future regarding immune systems-on-a-chip.

## 2. Physiology of the Immune System

The immune system of the human body is generally divided into two parts. The innate immune response includes defense mechanisms that are encoded in the host’s genes. These are typically physical barriers (e.g., the epithelial cell layers or mucus), soluble proteins, small molecules released from cells (e.g., cytokines or chemokines) or present in body fluids (e.g., complement proteins), membrane bound receptors and cytoplasmic proteins. The organs involved in this system are mostly secondary lymphoid organs, such as the spleen, tonsils, lymph nodes (LNs) and the cutaneous and mucosal organs. These are the organs where B- and T-lymphocytes recognize foreign antigens, initiate an effective immune response and help facilitate the crucial interactions between B- and T-cells. They contain zones with clusters of B- and T-cells [26]. Within these zones, dendritic cells (DCs) bind antigen–antibody complexes for efficient B-cell maturation and activation, as well as binding to antigen presenting cells for T-cell activation [27].

The adaptive immune system is programmed to be highly specific, based on unique antigen receptors located on the membranes of B- and T-lymphocytes. There are millions of lymphocytes, which cover a vast array of antigen receptors, all with unique specificities for certain antigens. The B- and T-lymphocytes mature in the BM and thymus, respectively, which are the body’s primary lymphoid organs [27]. They are activated upon contact with antigens that perfectly match their antigen receptors. When activated, these cells can differentiate into effector and memory cells by clonal selection. Effector cells are produced when the body is exposed to a certain antigen for the first time, and they defend the body during the primary immune response. Memory cells, on the other hand, are not activated during this phase. They activate upon exposure to the same antigen, initiating a quick and effective immune response to enable rapid neutralization of the pathogen [28,29].

However, other subsets of leukocytes and immune cells are also needed for a complete immune response. Pluripotent hematopoietic stem cells can differentiate and mature into B-cells, T-cells and natural killer (NK) cells. NK cells can identify virus-infected cells or tumor cells using their complex cell surface receptors [30]. Myeloid stem cells can differentiate into multiple granulocytes (e.g., monocytes, macrophages, neutrophils and eosinophils), megakaryocytes and erythrocytes. The granulocytes release large quantities of immunologically active molecules, such as reactive oxygen species (ROS) or enzymes, which can either kill invaders or activate other immune cells [31,32]. Macrophages are seen as one of the main regulators of the immune system. They can phagocytize microbial invaders, process their antigens and subsequently activate the adaptive immune response by presenting the antigens to T-cells. Depending on their phenotype, they can play important roles in both the pro- and anti-inflammatory processes by releasing either pro- or anti-inflammatory bioactive molecules [33].

The lymphatic vessel system is also specialized to effectively help the immune response. Specialized vessel structures, such as the endothelial venules in LNs and the marginal sinus in the spleen, efficiently lure naive B- and T-cells through afferent lymphatic vessels into a connected lymphoid organ containing the antigen information of the invading microbes by using chemokines or other specific signals (e.g., lysophospholipid sphingosine 1-phosphate). The efferent lymphatic vessels provide a fast way into the bloodstream for activated antigen presenting cells (APCs) [34,35].

Although these systems are often seen as independent, their synergy is crucial for an effective immune response. The innate immune system can activate antigen-specific cells, which become prominent after several days, after the cells have undergone clonal expansion. Moreover, the adaptive immune cells can recruit innate mechanisms to completely thwart the invading microbes [27].

The complexity of the immune system poses the biggest challenge in developing suitable biomaterials for applications such as steering the immune system. This complexity often results in adverse immune reactions, such as edema, pain, tissue destruction or total rejection of biomaterials [36]. Understanding biomaterial–immune system interactions is imperative for designing and developing new biomaterials. Immunomodulation and immunosuppression have been major research focuses over the last few decades, both of which have been comprehensively reviewed in the following studies [27,31,37,38]. Figure 2 provides an overview of the main (sub-)parameters in immunomodulation that are potentially regulated by the physical and biologic properties of biomaterials. Changing one of the subparameters, such as surface charge or shape, can have major consequences that are often unpredictable [39]. For example, Bartneck et al. showed that positively charged particles can lead to a higher activation of the immune system than negatively charged particles [40], which, consequently, inhibits immune function [41]. Yet, Kakizawa et al. provided some contradictory results, where negatively charged biomaterials induced a higher level of immune activation than positively charged ones [42]. Therefore, OoC models have become a progressively more prominent platform to study these parameters in a controlled environment. These microphysiological immune-system-on-a-chip models aim to emulate immune system organs, such as LNs and bone marrow (BM) and immune responses in a wide variety of tissues including liver, lung, skin or gut.

## 3. State-of-the-Art Immune-System-on-a-Chip Models

### 3.1. Lymph-Node-on-a-Chip

LNs are strategically placed throughout the body and play a key role in regulating the immune response. Immune cells in the LN, such as macrophages, cleanse the lymph fluid by filtering out microorganisms, pharmaceutical drugs and other foreign debris found in the interstitial fluid between tissues. Other cells, such as the B- and T-cells, activate the immune system when they encounter foreign antigens [43].

LNs are surrounded by a fibrous capsule and are divided into compartments by trabeculae, which are connective tissue strands that extend inward from the capsule. Furthermore, the LN consists of cortexes and medullas. The superficial part of the cortex houses follicles with germinal centers heavily packed with dividing B-cells. The deeper part of the cortex contains T-cells (e.g., cytotoxic (CD8) T-cells or memory (CD4) T-cells) in transit. DCs are abundantly present throughout the whole cortex, as they are critical for the activation and preparation of both B- and T-cells. The medullas consist of medullary cords, which contain both types of lymphocytes [43].

The complex architecture and organization make it difficult to mimic the LN in engineered tissues. Several groups have therefore set out to recapitulate specific areas or functions of the LN. Giese et al. developed so-called human artificial LNs (HuALN; Figure 3Ai,ii) that focused on the relationship between innate and adaptive immunity, the recognition of pathogens within the LN and the development of T-cell responses in vitro [44,45,46]. They designed a membrane-based perfusion bioreactor system containing: (1) an area for antigen-induced B-cell activation and dendritic cell (DC)–T-cell crosstalk supported by perfusable microporous hollow fibers, (2) a peripheral fluidic space to mimic the lymphatic drainage, and (3) a 3D hydrogel matrix loaded with DCs within two perfusable matrix sheets. Both B- and T-lymphocytes continuously moved from the peripheral fluidic space towards the DCs, which contained a 3D hydrogel matrix that would search for a specific receptor fit. Upon immunization, cell proliferation and antigen-dependent cytokine release, the formation of lymphoid follicle (LF) structures and a controlled release of antibody (IgM) production was found. Recently, this bioreactor was further studied using mesenchymal stromal cells [47].

Rosa et al. investigated the interaction between DCs and T-cells at varying shear stresses in a LN-on-a-chip (LNoC) flow device, which represented the paracortical region of LNs (Figure 3Bi,ii) [48]. The chip was fabricated from polydimethylsiloxane (PDMS) and consisted of one main flow channel with two inlets and two outlets that were bonded to a glass slide. The LN tissue was emulated by adhering an antigen presenting DC monolayer (activated by lipopolysaccharide (LPS), which is a highly proinflammatory molecule that is secreted by certain pathogenic bacterial cells and ovalbumin peptides presented by the major histocompatibility complex (pMHC)) in the main channel. T-cells (CD8^+^ and CD4^+^) were introduced into the chip at different flow rates and shear stresses. The immune cells were shown to exhibit different durations and strengths of cell interactions in a shear-stress-dependent manner. DCs were shown to have stronger interactions with CD4 cells than CD8 cells. Moreover, a more stable DC–T-cell interaction was found in the presence of specific antigens than unspecific antigens. This LNoC model thus allowed for the investigation of pMHC–T-cell receptor bonding mechanisms under controlled microenvironmental conditions. More studies have been performed to evaluate DC–T-cell interactions using LNoC models that were introduced in Table 1 [49,50].

A recent study by Goyal et al. [51] aimed to recapitulate the LN (lymphoid) follicle structure and function in order to study the class switching of B-cells, plasma B-cells differentiation and antibody production. The top channel of a microfluidic channel, which was separated by a membrane, was filled with media, whereas the bottom channel was filled with a high density of B-cells and T-cells in Roswell Park Memorial Institute (RPMI) medium, Matrigel and collagen type I, representing the extracellular matrix (ECM). The cells in the bottom channel were activated via the perfusion of cytokines, antibodies, *Staphylococcus aureus* Cowan I (SAC) and Fluzone. In the LNoC, the self-assembly of 3D LFs was shown along the entire bottom channel under perfused conditions. This indicated that flow and shear stress could induce and orchestrate LFs assembly. Within the LFs, the formation of clusters of plasma B-cells was shown after seven days of stimulation, which did not occur in 2D cultures. Moreover, class switching of B-cells was shown in the chip after stimulation with specific cytokines and antibodies (IL-4 and anti-CD80, respectively). Influenza vaccine (e.g., Fluzone), via antigen presenting DCs, was introduced into the hydrogel. Fluzone exposure resulted in increased levels of antigen-specific antibodies and the formation of plasma B-cells five days after immunization. Moreover, the human LN chip exhibited cytokine profiles similar to the human volunteers.

### 3.2. Bone-Marrow-on-a-Chip

The microenvironment of the BM is very intricate and is therefore difficult to replicate in vitro. The BM gives rise to hematopoietic stem cells (HSCs), which are capable of differentiating towards a plethora of immune cells after forming common precursor cells [16]. Recapitulation of the BM requires cellular, physical and chemical cues, engineered to maintain hematopoietic function. The first BM-on-a-chip was created by Torisawa et al. [52]. A cylindrical PDMS device was implanted in the BM of mice, together with osteogenic factors such as bone morphogenetic protein 2 (BMP2). After eight weeks, the PDMS device was successfully explanted and the formation of BM within the device was confirmed. To avoid adipocyte migration, which would inhibit BM function, the central cavity of the implanted device was closed by a solid layer of PDMS. The cell content was characterized, and HSCs and hematopoietic progenitor cells were observed inside the BM-on-a-chip [52]. The hematopoietic niche cells included osteoblasts, endothelial, perivascular cells and nestin^+^ mesenchymal stem cells (MSCs), and they were found in physiological positions in the device. The presence of nestin^+^ cells that support HSCs function and pluripotency [52,53] in the BM-on-a-chip suggested that the device could maintain HSC and hematopoietic function in vitro. The in vivo engineered BM (eBM) was then maintained in in vitro conditions within a microfluidic device. The researchers showed that the maintenance of the BM and its cellular functions lasted for up to seven days, offering a sufficient time window for investigating the efficacy and cytotoxicity of drugs. Remarkably, they showed that the culture medium did not require expensive cytokines to maintain the cellular function of the eBM [52]. Later, the BM-on-a-chip was used to study myeloerythroid toxicity after exposure to drugs and ionizing radiations [54]. In conclusion, a working model of a BM-on-a-chip was created, which allowed for real time monitoring of growth factor and cytokine secretion and drug testing/toxicity; however, it did not completely overcome the use of animals to study BM function.

A work conducted by Chou et al. [55] recapitulated BM hematopoiesis as well as BM dysfunction using a microfluidic chip. The device consisted of a top channel with primary BM stem cells and CD34^+^ progenitor cells seeded in a hydrogel and a bottom vascular channel with an endothelial cell lining. It was able to mimic hematopoiesis, as different blood cell lineages differentiated and matured, including neutrophils, erythroids and megakaryocytes, and it could maintain CD34^+^ cells for up to four weeks. Moreover, BM dysfunction was modeled using CD34^+^ from a source with a genetic disease (Shwachman–Diamond syndrome), which would form the same abnormalities of neutrophils as found in vivo. Therefore, this model can facilitate fundamental research on BM pathology and drug discovery. However, the presence and maintenance of HSCs, a key aspect of BM function, was not demonstrated. Additionally, research on the translation of other BM-related diseases should be conducted to show the full potential of the device in recapitulating dysfunctional BM of various origins.

A different BM-on-a-chip model was created by Sieber et al. [56]. They cultured primary human MSCs and umbilical cord-derived hematopoietic stem and precursor cells (HSPCs). The MSCs were precultured on a ceramic scaffold, allowing for ECM formation, which further allowed HSPCs to maintain their phenotype after being added to the culture system (Figure 3C). Upon cellular analysis, the researchers found the nestin^+^ expressed MSCs which promoted HSPCs to maintain their phenotype for up to four weeks. Other genes involved in hematopoietic niche functions (e.g., adhesion, vascular development, HSPCs chemotaxis and maintenance) were observed, which corresponded with the HSCs’ phenotype. The developed microfluidic device included a compartment dedicated to the BM, while another compartment could be used for a different organ to enable its interaction with the immune system (see Figure 3C–E). Bruce et al. [57] developed a 3D microfluidic chip for a triculture pathophysiological model to study acute lymphoblastic leukemia (ALL), a type of cancer affecting the blood and BM. This model included human bone marrow stem cells (BMSCs), an ALL cell line and human osteoblasts loaded in a collagen hydrogel, which were jointly placed in the microfluidic system. The researchers used this model to study the therapeutic efficacy and chemoresistance of chemotherapeutics.

In this recent research, despite the intricate microenvironment of the BM, organotypic elements have been successfully recapitulated on-chip, and this was shown to be sufficient to maintain HSCs’ phenotype by including nestin^+^ cells within the microfluidic chips. However, many technical challenges still remain before the microenvironments of native BM can be replicated without the use of in vivo engineering approaches.

### 3.3. Other Immune Organs

Compared to the LN and BM, the development of OoC models for other immune organs, e.g., tonsils, thymus, spleen, etc. has lagged behind. Regardless, some 3D in vitro models have become available for these immune organs. For example, the tonsils provide cues for the differentiation of plasma cells [58]. Tonsil organoids have been used to assess antigen-specific B- and T-cell responses [59]. The thymus is a primary immune organ, essential for the development of T-lymphocytes, and has been modeled with the use of organoids as well [18,60,61]. Moreover, the function of the thymus was recapitulated with the use of synthetic scaffolds and decellularized ECM, in combination with thymic epithelial cells [18]. However, a tonsil-on-a-chip or a thymus-on-a-chip have yet to be developed.

The largest secondary lymphoid organ in the human body is the spleen. Its immunological functions vary from clearance of red blood cells (RBCs) to hematopoiesis [62]. The spleen has two functional compartments: (1) the red pulp, containing the blood, which removes pathogens and cellular debris and (2) the white pulp, containing the lymphocytes, which initiates adaptive immune responses [63]. Moreover, the spleen is involved in the pathophysiology of malaria [64], sickle-cell anemia [65] and hemolytic anemia [66]. The pathology of these diseases is mostly understood based on animal models, in vitro studies and post-mortem examinations. However, the anatomy of the human spleen greatly differs from a rodent spleen [67,68]. In order to efficiently study the immune response in vitro, a human spleen-on-a-chip model holds great promise.

Buffet et al. [67] created the first ex vivo model of the spleen to study various pathophysiological conditions, including malaria. The spleen was surgically retrieved from the body and perfused ex vivo, allowing for vascular flow, metabolic activity and maintenance of the structure. Rigat-Brugarolas et al. [68] established a so-called human splenon-on-a-chip, modeling the red pulp of the spleen. The model was a microfluidic device, which accurately represented the blood flow within the red pulp, thereby mimicking the hydrodynamic behavior and filtering function of the spleen. It was shown that non-infected reticulocytes were less deformed by the chip than infected reticulocytes, proving the filtering mechanism of the device. The device could be used to recognize different types of RBCs. Regardless, only a few studies have reported the development or use of spleen-on-a-chip devices, especially for the white pulp compartment of the spleen. Future research could focus on understanding the immune function of lymphocytes in the spleen to gain insights into the diseases such as malaria.

## 4. Integration of Immune Cells and Components for Organs-on-a-Chip

### 4.1. Inflammation-on-a-Chip

The migration of immune cells is a hallmark for inflammation. Moreover, the interactions between immune cells, such as neutrophils, and endothelial cells are important, as this migration is regulated by the vasculature [69]. Transwell assays are classically used to study the migration of cells in 2D, but this does not accurately represent the complexity of the in vivo situation [70]. Therefore, researchers have started using microfluidic techniques including OoC platforms. Han et al. [71] used an inflammation-on-a-chip device to show the transendothelial migration of neutrophils, which can be used as a disease model of inflammatory diseases. As the chip only requires a limited amount of cells and is easy to manufacture, it could potentially be used for high-throughput drug screening. Ingram et al. [72] also modeled the migration of neutrophils in an inflammation-on-a-chip using induced pluripotent stem cell (iPSC)-derived endothelial cells. This model secreted angiogenic factors and inflammatory cytokines and is thus a potentially physiologically relevant model. Jones et al. [73] modeled the leukocyte migration using an inflammation-on-a-chip device. The immune cells were exposed to a gradient of proinflammatory chemoattractants and offered single-cell resolution analysis. This device offers the possibility of studying the fundamental mechanisms of inflammation, as the researchers found some unexpected results of leukocyte trafficking, indicating that the interactions between monocytes and neutrophils were more complex than would be expected based on our current understanding. Single-cell analysis was also enabled by Hamza et al. [74], who monitored neutrophil trafficking in response to chemoattractant gradients. Migration patterns of neutrophils, before and after phagocytosis, were mapped, which showed similar results to previous research. The behavior of the neutrophils before the monocytes arrived at the site of inflammation showed similar results as well. In future work, more fundamental research on the migration could be done, as well as possible uses the device could have for clinical applications.

The model used by Gopalakrishnan et al. [75] also used a gradient of proinflammatory cytokines to model inflammation (Figure 4A). In their model, macrophages, T-cell hybridomas, and DCs could move freely in the network of bifurcated microchannels in the device. The migration of immune cells could be followed in real time in response to exposure to a chemoattractant gradient, with similar migration speeds found in vivo. On-chip interactions were possible with this device, making it suitable for immunotherapy testing or cancer metastasis. However, the cells were not in a physiologically relevant environment, which is a limitation of the device.

Sasserath et al. [76] developed a three-OoC device consisting of the liver, cardiomyocytes and skeletal muscle, including the innate immune system, represented by circulating THP-1 monocytes. Its functionality was maintained for up to 28 days. Inflammation was induced through LPS and IFN-γ treatment and the cytokine response, as well as cell damage, were monitored. The system showed non-selective damage to the cells in the three different organs, as well as a secretion of proinflammatory cytokines, indicative of M1 polarization of the THP-1 monocytes. Moreover, upon treatment with cardiotoxic agent amiodarone, an increase in the anti-inflammatory cytokine IL-6 was recorded, suggesting that M2 polarization of the THP-1 monocytes occurred. Therefore, the microfluidic device allowed for the monitoring of intricate cell–cell and cell–immune interactions for three different organs, in addition to the observation of macrophage polarization. Moreover, the system used serum-free medium, as FBS is known to influence immune responses. However, the limitations of the chip include its monoculture of cell types which represented full organs and a full immune response.

Benam et al. [77] developed a microfluidic model of inflammation in the lungs and used it to monitor drug responses in vitro. The small airway OoC contained a bronchiolar epithelial layer with a functional mucosal layer, as well as a vascular endothelial layer. They induced an inflammatory response as a result of an IL-13 insult, which displayed hyperplasia of the mucus secreting goblet cells, in addition to a proinflammatory cytokine response. Moreover, they used epithelial cells derived from individuals suffering from chronic obstructive pulmonary disease (COPD), which modeled the pathophysiological situation depicted by neutrophil recruitment. The device presents an attractive method for drug screening and for the identification of biomarkers in inflammatory pulmonary diseases, however, it lacks a prolonged inflammatory response, including the recruitment of macrophages and other immune cells, as a continuation of the disease.

To date, most inflammation-on-a-chip devices have used a gradient of proinflammatory cytokines to study migration patterns. These inflammation-on-a-chip models could be used for evaluating drugs or for immunotherapy. However, other types of studies using pathologic scenarios have remained scarce and there is also still no comprehensive understanding of inflammation. Therefore, it is important to fundamentally study the migratory effects of inflammation via pathologic scenarios. Performing such studies could allow for a better understanding of the drug and increase the chances of a clinical success. Moreover, the reader can be referred to the review of Irimia et al. for an overview of in vitro techniques to study the immune system and inflammation [78].

### 4.2. Skin-on-a-Chip

Similar to the gut lining, the skin is a first physical line of defense to protect the body against pathogens. However, the skin does not offer protection only in the form of a physical barrier, but it also functions as an active immune organ. The epidermis contains keratinocytes, which express toll-like receptors to distinguish pathogens that secrete cytokines upon an encounter and activate the immune cells in the dermis, which contains DCs and T-lymphocytes [79]. Ramadan et al. [80] created an immune-competent skin-on-a-chip model by co-culturing immortalized keratinocytes (HaCaT) and a human leukemic monocyte lymphoma cell line (U937), which was chosen as an alternative to DCs (Figure 4B,C). Of both cell lines, the monoculture and the co-culture model were subjected to LPS treatment, and the expression of proinflammatory cytokines IL-1β and IL-6 was measured. This resulted in the highest expression of inflammation in the monoculture of the U937 cells. The tight junctions of the skin were improved by dynamic perfusion of the platform, recapitulating the in vivo situation. The chip remained functional for up to 17 days, allowing the researchers to investigate the effects of toxicological studies. Moreover, the chip could potentially be used for a variety of applications, including cosmetics as well. This OoC is currently the only microfluidic platform that encompasses the immune function of the skin, however, the use of (cancerous) cell lines limits its physiological relevance.

Wufuer et al. [81] established a skin-on-a-chip with inflammation based on exposure to TNF-α, which corresponded to pathological skin inflammation. The skin-on-a-chip consisted of three layers, corresponding to the three layers of the human skin. The secretion of IL-1β, IL-6 and IL-8 were monitored, similarly to the gut-on-a-chip from Kim et al. [82] and could be used for drug testing. Other research on skin-on-a-chip platforms indicated the potential for the incorporation of immune cells [83], which are anticipated in future research.

### 4.3. Liver-on-a-Chip

The liver contains the largest group of resident macrophages, namely the Kupffer cells, which play a major role in the secretion of inflammatory cytokines and the production of complementary components in the immune response [84]. Therefore, the liver could offer a great potential for the integration of immune-cells-on-a-chip, since it is useful for studying the interactions between the drugs and the immune cells, as almost all drugs pass the liver and could accelerate drug development [85]. It would be ideal to study drug metabolism on a liver-on-a-chip device, specifically one which models the liver metabolism on a chip [86,87]. Moreover, the interactions of drugs with the immune system (and immune cells) should ideally be studied in a controlled in vitro environment. Unfortunately, there are very few reports on the interactions between the liver and immune cells in 3D in vitro models. A liver-on-a-chip device developed by Emulate, Inc. may facilitate the incorporation of an immune component [88], but this has not been reported thus far. However, there is a study on liver-on-a-chips that reports on the interaction between liver tissue and monocytes, which mimic liver inflammation [89]. In this system, the main cell types of the liver, namely the hepatocytes, hepatic stellate cells, endothelial cells and primary macrophages, an alternative to Kupffer cells, were used (Figure 4D). The migration and polarization of monocytes were evidenced upon LPS treatment. The LPS-induced M1 polarization, which is classically seen as proinflammatory. However, upon monocyte invasion, the cells produced IL-10, which is an anti-inflammatory cytokine, leading to anti-inflammatory M2 polarization. Although the platform contained all four major cell types of the liver, it did not take the structural positioning of the tissue nor the relative proximities between cells into account, which would likely limit the device’s ability to emulate native cell–cell interactions. To overcome these technical shortages, more research should be performed in the near future for replicating the metabolic activities and immune responses of native liver tissue.

### 4.4. Gut-on-a-Chip

The gut is a highly important organ for the immune system, as the gut lymphoid tissue and immune system directly interact with the microbiome. Remarkably, these interactions have not been studied extensively on an OoC platform [91]. Moreover, the lining of the gastrointestinal tract is subjected to the external environment and requires immune function, which is achieved by the intestinal mucus layer. In order to study the immune interactions in the gut, various culture models have been used, ranging from a 2D Caco-2 cell layer interacting with THP-1 monocytes [92] to intestinal organoids with microbiota niches [93]. Ideally, intestinal models should include an epithelial layer, immune component, peristaltic motion, microbial interactions, mucus and transport of nutrients [16]. In particular, the peristaltic movement of the gut is an important organotypic function that should be recapitulated in vitro. Kim et al. [82,90,94] made a peristaltic gut-on-a-chip device with two channels, separated by a porous PDMS membrane coated in ECM. Both sides of the membrane were lined with intestinal epithelial cells, forming villi after five days, and the device could be successfully cocultured with microbial cells. The channels were placed in between two vacuum chambers, which allowed for the recapitulation of the peristaltic movements (Figure 4E) [90]. Subsequently, the chip was used to mimic inflammation by coculturing the gut-on-a-chip with a pathological strain of E. coli, which causes extreme diarrhea and human lymphatic microvascular endothelial cells on the other side of the PDMS membrane [82]. This resulted in the secretion of proinflammatory cytokines such as TNF-α, IL-1β, IL-6 and IL-8, indicating an inflammatory response. Kim et al. found that there was an overgrowth of bacteria, which resulted in epithelial deformation and disturbance in the peristaltic movements, as seen in patients suffering from chronic inflammatory bowel diseases, such as Crohn’s disease [82]. The gut-on-a-chip could therefore potentially be used to study the pathophysiology of the disease, as well as drug delivery and drug interactions with the immune system. However, the system currently includes only a small number of cell types and thus does not encompass all required immune cell interactions, i.e., macrophages, which represents the key limitation of this chip.

Another gut-on-a-chip that showed interactions with the immune system was developed by Shah et al. [95]. Using two independently controlled channels, they were able to co-culture human epithelial cells aerobically and microbial cells anaerobically in a single microfluidic device. Moreover, primary human CD4_+_ T cells were cultured in the chip without any loss of viability for either of the cellular components. This device could thus be used to examine the gut’s response to drugs, in particular, the subsequent interactions with the immune system, as well as more a fundamental understanding of gut-microbe interactions. Other gut-on-a-chip systems have the potential to incorporate an immunocomponent, e.g., the human gut epithelial cell line Caco-2 has shown a response when co-cultured with immune responsive cells (U937) [96]. Various types of gut-on-a-chip systems integrated with immune-components have been established and studied. However, new research for chronic inflammation in the gut or other immune related systems in the gut, should be further performed on a device with peristaltic motions and which incorporates an epithelial layer of, preferably, iPSCs and immune cells.

### 4.5. Tumor-Microenvironment-on-a-Chip

The tumor microenvironment (TME) is complex, with interactions between malignant cells, non-malignant cells, immunomodulatory cells and the ECM. Three facets within the TME are important: (1) the hypoxic core which controls metabolic shifts of the cancer cell niche, (2) induction of angiogenesis by the tumor and tumor stroma and (3) the interactions of the cancer cells with the stroma and the immune system. Cancer-associated inflammation contributes to cancer cell proliferation, genomic instability, stimulation of angiogenesis, cancer antiapoptotic pathways and cancer dissemination. Thus, more research has focused on the fabrication of a tumor-microenvironment-on-chip (TMoC) to mimic and understand the intricate relationship between the TME and the immune system [97,98].

Parlato et al. designed a novel microfluidic system to monitor the behavior of patient-derived interferon-alpha-conditioned DCs (IFN-DCs) towards colorectal cancer cells (CRCs), which were either untreated or treated with a novel antitumor treatment containing romidepsin and IFN-α2b (RI) [99]. The real-time monitoring of the immune-tumor interactions allowed them to further reveal how this new RI cancer treatment could increase the antitumor functions of immune cells. Results showed the migration of IFN-DCs towards RI-treated CRCs resulted in an increase in the phagocytosis of the RI-treated CRCs (not observed in the untreated CRCs).

The role of oxygen in immunosuppression within the tumor microenvironments was assessed by Ando et al. [100]. They recapitulated the hypoxic tumor microenvironment inside a microfluidic chip to assess the cytotoxicity of chimeric antigen receptor T (CAR-T) cells against ovarian cancer cells (OCCs). CAR-T cells were delivered in microfluidic channels to the OCCs, which were embedded in gelatin methacryloyl (GelMA) with oxygen-diffusion-barrier pillars to create an internal oxygen gradient. It was shown that hypoxia altered the programmed cell death-ligand 1 (PD-L1). Moreover, CAR-T infiltration was hindered due to matrix stiffness as well as oxygen concentration.

In another study, Ayuso et al. studied the interactions between NK cells and human breast tumor spheroids, which were co-cultured in a collagen gel [101]. NK cells and a spheroid laden channel were flanked by two endothelial cell lined vascular channels, which allowed for the perfusion of antibodies into the collagen gel. A modified antibody–cytokine conjugate, anti-EpCAM-IL-2, was used to bind to the epithelial cell adhesion molecule (EpCAM) protein expressed on the surface of the tumor cells. IL-2 also induced NK cell proliferation. As expected, the antibody penetration was delayed by the endothelial cell barrier, but, once penetrated, they could diffuse throughout the matrix. Subsequently, the penetration of the antibodies was limited to the periphery of the spheroids due to the cell–cell interactions. However, this enhanced the infiltration and proliferation of NK cells, leading to NK-cell-mediated cytotoxicity, higher antitumor efficiency and the destruction of the spheroid.

Cancer metastasis is the primary cause of cancer-related deaths and is thus one of the main research topics within TMoCs. Initially, most studies have focused on the mechanisms of tumor metastasis via the bloodstream. Yet, most metastatic cancers spread via lymphatic vessels to draining LNs and end up in distant organs [102,103,104]. Thus, more biologic knowledge behind lymphatic metastasis is required, since it remains ambiguous how cancer cells alter the genetic profiles of lymphatic endothelial cells and how these contribute to lymphatic dysfunction. To study this in more detail, Ayuso et al. developed a lymphatic microfluidic model with estrogen-receptor positive (MCF-7) and triple negative (MDA-MB-231) breast cancer cells and human lymphatic endothelial cells [105]. A breast-cancer cell-filled lumen adjacent to a tubular lymphatic vessel was co-cultured in a collagen hydrogel. The results revealed that the genes involved in vessel growth, permeability, metabolism, hypoxia and apoptosis were altered in the cells co-cultured with MCF-7 cells. This also led to functional changes in the endothelial barrier functions, such as lymphangiogenic sprouting and higher permeability to 70 kDa dextran and glucose [105].

Table 1 provides an overview of other studies on tumor-microenvironments-on-chips with an immune component. These studies highlight the importance of developing more advanced models to understand the most important parameters, i.e., immune system, in the delivery of therapeutic agents. However, these models could still be improved in the future to mimic a more relevant tumor microenvironment more precisely. For example, the incorporation of organotypic endothelial cells derived from a specific organ or stromal cells, should increase the fidelity of the native and physiological tumor microenvironment. Moreover, the reproducibility and robustness of the chips are still complications that should be overcome.

## 5. Limitations and Future Perspectives

This review provides a comprehensive overview of the current state-of-the-art immune-OoC platforms, as well as OoC devices with immunofunctionality. In addition to the key advantages of these platforms, we have also highlighted the current drawbacks, which could fuel future research. Despite the increasing research into LNoCs, only a few studies have tried to mimic one of the most important functions of the LN, namely its ability to recognize and fight pathogens and infections. Instead, most research has focused on chemotaxis and immune response to chemokine gradients within lymphoid structures, which are undoubtedly significant functions. However, mimicking the antipathogenic functions of the LN on a microfluidic platform is of vital importance in drug development. Moreover, no microfluidic platform until now has been able to reproduce the full function of the LN, partially due to its complex architecture. LNs are characterized by a large variety of cells, continuous migration of immune cells and dynamic cell–cell and cell–ECM interactions [111]. Achieving such a dynamic and complex structure in vitro is not a trivial challenge. More research into biomaterial scaffolds and their physical and chemical properties (Figure 2) would provide a step forward in more accurately replicating the dynamic in vivo structure of lymphoid structures.

Similar to the LNoC, research in BM microfluidic devices has encountered several limitations. For example, the first and most complete BM model by Torisawa et al. [52] still required in vivo engineering of the tissue, whereas OoC devices could be used to reduce the use of experimental animals. Regardless, this approach does recapitulate the in vivo BM situation well. Unfortunately, the device could only be used for up to seven days. No non-acute cytotoxic reactions (over seven days) could be tested on this device, which limits the applications of the device. The BM-on-a-chip by Sieber et al. [56] had an increased culture period of up to four weeks, but this device used a ceramic to mimic the structure of the BM tissue, which could lead to detrimental effects.

Several models within the field of TMoC have been progressively more able to mimic and define important parameters for the delivery of therapeutic agents. However, these models could still be further improved via the use of organotypic cells. Nowadays, most chips are dependent on cells derived from mice, rats, or cell lines, as these cells are often easier to use and provide the researcher with a concept model. However, the addition of human organotypic cells could lead to an increase in the reliability and relevance of the TMoC.

For the other OoC models with immune functions, research has yet to progress to a relevant stage. In recent years, a few studies have been reported that focused on on-chip immune reactions for certain organs, which could pave the way for a novel standard in drug discovery and pharmaceutical evaluations. Most OoCs poorly recapitulate the in vivo microenvironment, owing to their reliance on immortalized or cancerous cell lines. Current OoCs can be improved by using primary cells instead of tumor cell lines and by increasing their lifetimes to study long term immune responses. Additionally, the engineered organ’s compositional and spatial structure could be improved to resemble the natural complexity of its native counterpart more closely. Moreover, the immune response is mostly mimicked by inflammatory cytokines and chemokines and lacks the use of immune cells such as macrophages. It would be interesting to monitor the polarization of the macrophages into M1 and M2 to model the immune response in specific organs. Moreover, none of the discussed microfluidic models are able to perform a simultaneous high-throughput screening of (a combination of) many inflammatory factors (e.g., drugs, cytokines, chemokines, etc.). This would require the use of multiple immune cells, all cultured on a single chip, either in separate or connected chambers and channels. The fact that some studies [44,45,46,51] have shown the incorporation of several inflammatory factors indicates that these devices could possess this potential.

Most state-of-the-art models recapitulate the healthy physiology of the (immune) organ. There are very few models which make the transition towards a diseased state to understand the pathophysiological state and to be able to identify potential new biomarkers, usable for the formulation of pharmaceutical solutions. The small-airway-OoC from Benam et al. [77], for example, made the transition from a physiological model to a pathophysiological model by using diseased cells in their microfluidic device. Other models can achieve a pathophysiological state by incorporating a similar strategy as well. Moreover, OoC platforms have aided in the shift from 2D to 3D systems by implementing more complexity and by mimicking the in vivo (immune) responses more closely. However, in some cases, there is no need to overcomplicate the device design. In those cases, simplifying the (patho)physiology is equally important as answering the relevant biologic questions, as shown by several studies [25,50,106,112]. Thus, researchers should be careful to avoid both oversimplifying the in vivo (patho)physiology, as well as overcomplicating the device design.

A great number of immunomodulatory OoCs use inflammatory biomolecules (i.e., pro- and anti-inflammatory cytokines) to modulate the cells to obtain the desired immune response. Due to the high costs of these molecules, the focus should shift towards the use of new generations of biomaterials (i.e., synthetic peptide structures and cell-responsive polymers [113,114]) or the use of cell–cell interactions [115,116]. This would also increase the throughput, reproducibility and possible upscaling of the system. Moreover, a common material used in many OoCs is PDMS. However, PDMS is highly lipophilic and binds molecules and drugs that are present in the perfusion medium. This effect can be reduced by modifying the PDMS or by using different immunomodulatory biomaterials or anti-fouling nanocoating [117]. This would also increase the reproducibility of the chip. In addition, there have been recent efforts to translate PDMS-based academic research to commercialized products using recently developed advanced fabrication techniques, such as injection-molding and 3D printing. Biocompatible resins mimicking desirable PDMS properties should be used to prevent any potential toxicity from the resin [118]. Therefore, a combination of these advanced microfabrication techniques and biocompatible resin could be used to form robust and reproducible microfluidic chips in an automated and high-throughput fashion, although those devices have not been used widely in immunology studies.

Until now, no multiplexed immune system-on-a-chip that incorporates multiple immune organs has been developed. Single organ chips cannot be reliably used to investigate a drug’s systemic effect or recapitulate the full immune response, as different organs are involved in the orchestration of a full immune response. A multiorgan chip should therefore have multiple cell types, all in a separate ‘organ’ chamber, which are connected via channels and should preferably be monitored in real time [119]. Despite the development of multiorgan devices [120,121,122], no model exists for a multi-immune-organ on a chip. Moreover, the research on the application of the immune-OoC for drug testing has remained scarce as of yet. Therefore, a multi-OoC should follow the normal administration routes of drugs, for example, through the stomach and gut into the bloodstream and through the liver or through the skin and bloodstream, passing the liver. For drug testing, it is especially important to develop a functional liver-on-a-chip with an immune component. In conclusion, research on the immune system has come a long way, starting with the interest of Ancient Greek philosophers, to the development of complex OoCs. The research has come long way, and it is predicted it will reach further beyond our imagination.

## Figures and Tables

**Figure 1 micromachines-11-00849-f001:**
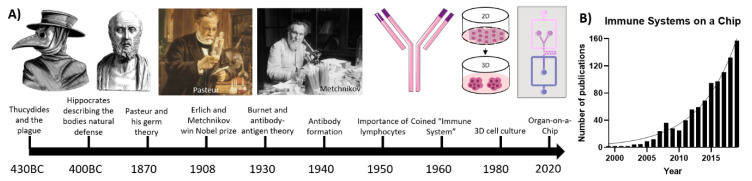
(**A**) Timeline of the important events in the history of immunology from 430 BC until 2020; (**B**) rapidly increasing publications for immune related microfluidic technologies and their applications in the biomedical engineering fields.

**Figure 2 micromachines-11-00849-f002:**
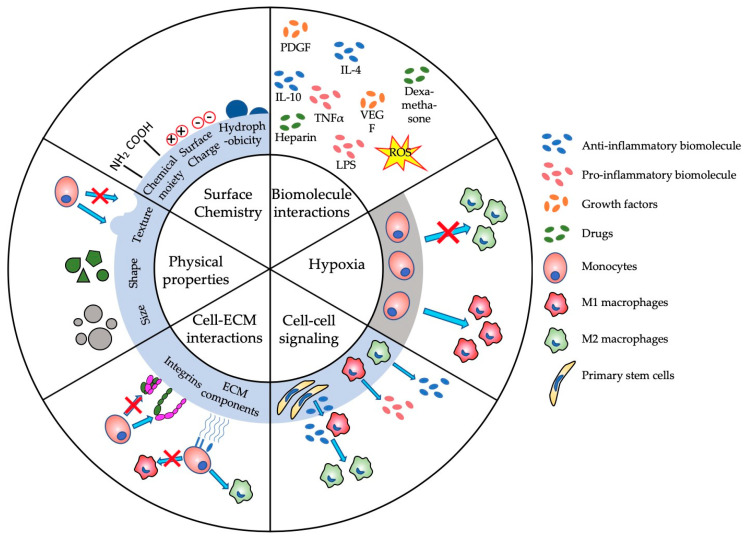
Schematic of major parameters that influence the immunomodulation of biomaterials and tissue engineered constructs. ECM—extracellular matrix; IL—interleukin; LPS—lipopolysaccharide; PDGF—platelet-derived growth factor; TNFα—tumor necrosis factor α; VEGF—vascular endothelial growth factor.

**Figure 3 micromachines-11-00849-f003:**
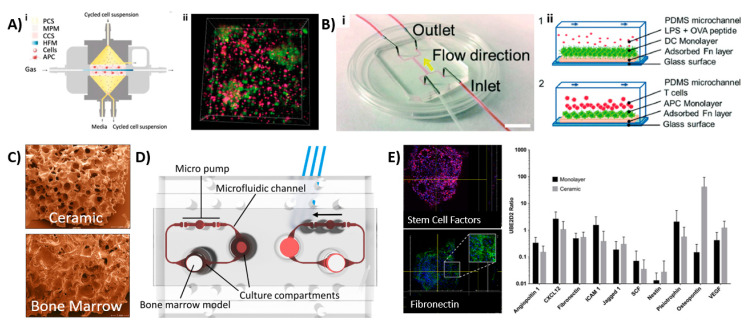
Overview of various LNoC and BM-on-a-chip models. (**A i,ii**) Schematic of the bioreactor, where cell culture media and cell suspensions flow vertically, and the gas supply perfuses horizontally (left). Formation of microorganoids of DCs (red), B cells (red) and PBMCs (pink) after 7 days of perfusion [44]. Reproduced from Ref. [44] with the permission from Elsevier. (**B i,ii**) A LNoC composed of one main flow channel with 2 inlets and 2 outlets (left). Cross sectional view of a schematic showing the DC–T-cell interactions during flow. 1—Dendritic monolayer with LPS or OVA activation to mimic the inflammatory response, 2—T-cell loading and interactions with DCs. (right) [48]. Reproduced from Ref. [48] with permission from The Royal Society of Chemistry; (**C**–**E**) BM-on-a-chip from Sieber et al. [56]. Copyright 2017 Wiley. Used with permission from Sieber et al., Bone marrow-on-a-chip: Long-term culture of human hematopoietic stem cells in a three-dimensional microfluidic environment, Journal of Tissue Engineering and Regenerative Medicine, Wiley [56]; (**C**) scaffold used to culture the MSCs (top) compared with human BM (bottom); (**D**) schematic figure of the BM-on-a-chip with possibilities for other organ implantation, resulting in a multiorgan-on-a-chip device; (**E**) immune function of the device, depicting the stem cell factor immunoexpression and fibronectin immunoexpression, as well as the expression of certain BM markers and BM niche cells, such as nestin^+^ and osteopontin. APC—antigen presenting cells; CCS—central culture space; DC—dendritic cells; HFM—hollow fiber module; LPS—lipopolysaccharide; MPM—microporous membrane; OVA—ovalbumin; PCS—peripheral culture space; PDMS—polydimethylsiloxane.

**Figure 4 micromachines-11-00849-f004:**
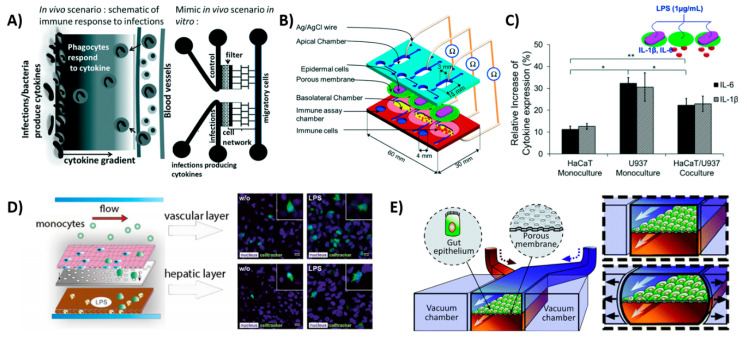
Schematic views of OoCs with immune cell components. (**A**) Inflammation-on-a-chip by Gopalakrishnan et al. [75] mimicking the cytokine gradient in a normal wound and the migration of immune cells. Reproduced from Ref. [75] with permission from The Royal Society of Chemistry. (**B**) skin-on-a-chip by Ramadan et al. [80] with dermal and epidermal layer, showing the interactions of DCs with the keratinocytes and (**C**) showing the expression of inflammatory cytokines IL-6 and IL-1β in the device with a keratinocyte (HaCaT) monoculture and a dendritic cell (U937) monoculture and the co-culture. Reproduced from Ref. [80] with permission from The Royal Society of Chemistry; (**D**) liver-on-a-chip by Gröger et al. [89]; the top layer consisting of endothelial cells and macrophages and the bottom layer consisting of hepatocytes and hepatic stellate cells. LPS is used to induce an inflammatory response and monocytes start migrating; (**E**) gut-on-a-chip by Kim et al. [90], showing the peristaltic motion as a result of the two vacuum chambers. This chip model is used to incorporate immune cells in later stages of the research [82]. Reproduced from Ref. [82] with permission from The Royal Society of Chemistry.

**Table 1 micromachines-11-00849-t001:** Engineering tumor immune microenvironments.

Design of Microfluidic Device	Key Cell Types	Findings	Ref.
**Lymph-node-on-a-chip**			
Membrane-based perfusion bioreactor system containing multiple chambers for antigen-induced B-cell activation, DC–T-cell crosstalk, peripheral space to mimic lymphatic drainage and a DC-loaded hydrogel (Figure 3Ai).	B and T-lymphocytes from healthy donors Monocyte-derived human DCs	Migration of B- and T-cells from peripheral fluidic space towards DCs. LF structure formation upon immunization and activation. Controlled IgM release post-activation.	[44]
Membrane-based perfusion bioreactor system containing culture compartment with LN cells and MSCs -laden agarose gel discs.	Rat-derived MSCs LN cells derived from rat lymph nodes	Concanavalin A-stimulated LN cells showed reduced proliferation in MSC co-culture. MSC co-culture suppressed levels of proinflammatory molecules (TNFα and IFNγ) and induced IL-1a and IL-6 secretion.	[47]
Two chamber microfluidic system with recirculating flow to transport secreted signals between tumor and lymph-node tissue.	BALb/c-derived tumor and lymph node tissue slices	Real-time monitoring of tissue interactions, fluid flow and shear stress. Decreased IFNγ secretion within lymph nodes cultured with immunosuppressed T-cell containing tumor tissue	[50]
PDMS chip with one flow channel connected to two inlets and two outlets.	LPS-activated DCs CD8^+^ and CD4^+^ T-cells	Duration and strength of immune cell response depended upon shear stress. Stronger DC interaction with CD4^+^ T-cells.	[48]
Microdevice with chemotaxis compartment filled with DCs linked to a T-cell compartment. Separate media and chemokine channels.	MUTZ-3-derived DCs, T-lymphocytes	Design allowed chemotaxis of DCs under non-adherent conditions. CCR7-induced mature DC migration towards T-cells. Mature DCs showed stronger T-cell activation than immature DCs. Showed chemotaxis is critical in T-cell activation.	[49]
Two-channel device with media in upper channel and B- and T-cells laden Matrigel in bottom channel.	B-lymphocytes, T-lymphocytes	Perfusion stimulated the formation of LFs inside the chip. Formation of plasma B-cell clusters 7 days post-stimulation. Class-switching of B-cells was induced with specific cytokines and antibodies. Similar cytokine profiles were observed to human volunteers when exposed to Fluzone.	[51]
**Bone-marrow-on-a-chip**			
Cylindrical PDMS device suitable for implantation.	HSCs Hematopoietic progenitor cells Osteoblasts Endothelial cells Perivascular cells Nestin^+^ MSCs	Formation and characterization of BM within device 8 weeks post-implantation. Presence of nestin^+^ cells indicate support of HSC and hematopoietic function. No expensive cytokines were needed to maintain cellular function.	[52]
Microfluidic chip device with central chamber containing BM tissue with underlying microfluidic channel, separated by a porous PDMS membrane.	In vivo-derived BM tissue	BM tissue produced and released blood cells into microfluidic circulation. Able to maintain viability and function of HSCs, which could differentiate into mature blood cells on-chip. Organ-level response to radiation toxicity. Showed that the hematopoietic microenvironment is crucial for modeling radiation toxicity.	[54]
Two-channel device with BM stem cell- and CD34^+^ progenitor cell-loaded hydrogel in top channel and endothelial cell lining in bottom vascular channel.	BM stem cells CD34^+^ progenitor cells Endothelial cells	Differentiation and maturation of different blood cell lineages, including neutrophils, erythroids and megakaryocytes. Maintain CD34^+^ viability up to 4 weeks Successful modeling of BM dysfunction using diseased CD34^+^ cells.	[55]
Microfluidic device consisting of a BM compartment and a compartment for other organs.	hMSCs HSPCs	Preculture of MSC on ceramic scaffold-induced ECM, which allowed maintenance of HSPC phenotype. Range of genes which are involved in multiple hematopoietic niche functions were observed.	[56]
Four-channel microfluidic platform filled with tumor cell, BMSC and HOB-laden collagen I.	Human Philadelphia chromosome positive B lineage ALL cell line BMSCs HOBs	Cell-matrix interactions influenced cell migration and invasion and led to cellular responses not observed in 2D. No BMSC spreading was observed in 3D dynamic condition. Decreased chemotherapeutic drug sensitivity was observed compared to 2D cultures.	[57]
**Splenon-on-a-chip**			
Two-layered microengineered device which mimicked the closed-fast and open-slow microcirculation.	Uninfected and infected red blood cells	Microfluidic device accurately mimicked the red pulp and thus the filtering function of the spleen with accurate recognition of different RBC types.	[68]
**Inflammation-on-a-chip**			
Multichannel device incorporating a co-culture of neutrophils and endothelial cells, ECM and concentration gradients of various inflammatory proteins.	Neutrophils Endothelial cells	The system showed transendothelial migration of neutrophils. N-formyl-methionyl-leucyl-phenylalanine showed higher attraction than IL-8. Strong correlation between matrix stiffness and migration was found.	[71]
Microfluidic culture platform with lumen channel inside a protein matrix.	Neutrophils iPSC-derived endothelial cells	Precise control over lumen size, structure and configuration. Composition of the ECM influences the barrier function of endothelial cells. Secretion of angiogenic and inflammatory factors. Neutrophil chemotaxis towards IL-8 improved in presence of endothelial cells.	[72]
Microfluidic device containing a central cell loading chamber and a chemoattractant gradient along migration channel.	Primary human neutrophils Human monocytes	Maintains chemotactic gradients up to 48 h but does change over time. Allows single-cell resolution of chemotaxis of neutrophils. Indication of bidirectional communication between monocytes and neutrophils.	[73]
PDMS device with central loading inlet, leading to eight channels connected to the chemoattractant chambers.	Human whole blood	Assay allowed passaging of neutrophils only. Chemoattractant gradients were maintained up to 8 h. Showed that neutrophils could regulate their traffic in absence of monocytes.	[74]
Multichannel PDMS device which allowed migration of cells through migration channels towards cytokine-laden channels.	MF2.2D9 T0 cell hybridomas IC-21 macrophages Immortalized B6 macrophages LPS-activated DCs	Successful migration of cells by chemoattractant gradient. Phagocytosis stopped macrophages from migrating further. Little cell proliferation observed. CCL19-induced mature DC chemotaxis.	[75]
A three-organ device with a liver module, cardiac cantilevers and stimulation electrodes, skeletal muscle cantilevers and recirculating THP-1 monocytes in medium.	THP-1 monocytes Primary human hepatocytes Human cardiomyocytes Human skeletal muscle myoblasts	Non-selective damage to cells in three different organs. Increased proinflammatory molecule release. Amiodarone-induced M2 polarization indicated by increased IL-6 release.	[76]
**Lung-on-a-chip**			
Two-channel device with a polyester membrane. Primary human airway epithelial cells cultured on membrane in upper channel, with medium flowing in bottom channel.	Primary human airway epithelial cells Epithelial cells derived from COPD patients.	Inflammatory response was induced by an IL-13 insult, resulting in a proinflammatory response with hyperplasia of mucus secreting goblet cells. Showed neutrophil recruitment to diseased epithelial cells.	[77]
**Skin-on-a-chip**			
Multilayer device with layer of HaCaT cultured on top of a porous membrane and an immune cell layer positioned beneath the KC layer.	HaCaTs U937 cell line	U937 monoculture showed highest expression of inflammation after LPS treatment. Perfusion induced the formation of tighter junctions.	[80]
Multilayer chip consisting of a HaCaT layer, a fibroblast layer and an endothelial cell layer, separated by porous membranes.	HaCaTs HS27 fibroblasts HUVECs	Successful design of skin model to mimic epidermis, dermis and vessels of the skin. Dexamethasone prevented tight junction damage and lowered IL-1β, IL-6 and IL-8 expression, thereby showing recovery of skin with edema.	[81]
Multichambered microfluidic device with interchangeable lids and insets for developing a full-thickness skin-on-a-chip model.	Human primary foreskin-derived dermal fibroblasts Immortalized human N/TERT keratinocytes	Developed a flexible bioreactor for tissue culture, with the ability to perform TEER measurements, permeation assays and assessing the skin’s integrity. Potential to culture multiple organs in parallel or addition of immune system. Dynamic perfusion improved morphogenesis, differentiation and maturation.	[83]
**Liver-on-a-chip**			
Multilayer biochip containing a HUVEC/macrophage layer with monocytes freely flowing in the media and a hepatocyte/hepatic stellate cell layer at the bottom.	HepaRG hepatocytes HUVECs LX-2 stellate cells Peripheral blood mononuclear cell-derived macrophages Primary monocytes THP-1 monocytes	Migration and M1 polarization of monocytes upon LPS treatment. IL-10 production upon monocyte invasion, inducing M2 polarization. Monocyte invasion inhibited inflammation-related cell death and induced the recovery of metabolic functions.	[89]
**Gut-on-a-chip**			
Two-channel device with porous membrane coated with ECM, with one side of the membrane coated with intestinal epithelial cells and the other with endothelial cells. Incorporation of vacuum chambers allowed recapitulation of peristaltic movements.	Caco-2 intestinal epithelial cells Human capillary endothelial cells. Human lymphatic microvascular endothelial cells E. coli strain	Formation of intestinal villi in 5 days. Inflammation was induced by co-culture with the E. coli strain, leading to secretion of TNF-α, IL-1β, IL-6 and IL-8 Growth of bacteria also resulted in epithelial deformation and disturbance of peristaltic movements.	[82,90,94]
Multichambered chip with separately controlled microbial and epithelial cell microchambers.	Caco-2 intestinal epithelial cells Noncancerous colonic cell line Primary CD4^+^ T-cells *Lactobacillus rhamnosus GG*	Successful incorporation of co-culture of human and microbial cells. Independently controlled chambers allowed for anaerobic culture conditions for the microbial cells. Slight inflammatory response after addition of microbial cells. Showed crosstalk between microbial and human cells, depicted by alteration of several genes and miRNAs.	[95]
Microfluidic device with apical and basolateral compartments separated by a porous membrane.	U937 cells Caco-2 intestinal epithelial cells	Full, confluent layers formed 5 days after Caco-2 cell seeding. Dynamic cell culture conditions improved viability. LPS and cytokine addition increased permeability of the epithelial cell layer.	[96]
**Tumor-microenvironment-on-a-chip**			
Central immune chamber with floating IFN-DCs connected to two side tumor chambers with treated and untreated cancer cells in type I collagen.	IFN-DCs RI+ and RI- SW620 CRCs	IFN-DCs migrated towards RI-treated cancer cells. Increased antigen take up resulting in increased phagocytosis and antitumor function.	[99]
CAR-T cells delivered through microfluidic channels. Tumor cells in GelMA between two oxygen diffusion barriers.	HER2+ SKOV3 human OCCs Anti HER2 CAR-T cells	Hypoxia alters PD-L1 expression. Limited CAR-T infiltration due to matrix stiffness and oxygen concentration. Hypoxia promotes immunosuppression.	[100]
Channel containing tumor spheroids embedded with NK cells in collagen. Two endothelial vascular lamina on lateral sides.	MCF7 breast tumor spheroids NK-92CD16V NK cells HUVECs	Delayed anti-EpCAM-IL-2 antibody penetration by endothelial barrier and cell–cell interactions. NK cell cytotoxicity and ADCC was enhanced by anti-EpCAM-IL-2.	[101]
Tubular lymphatic vessel adjacent to lumen filled with breast cancer cells, co-cultured in collagen hydrogel.	Estrogen-positive MCF-7 cells MDA-MB-231 breast cancer cells Human lymphatic endothelial cells (HLECs)	Co-culture with MCF-7 led to alteration of multiple HLEC genes, which correlated to functional changes in endothelial barrier capacity.	[105]
One channel filled with liver tumor cells in type I collagen. Second channel containing tumor specific T-cells. Control over oxygen levels and inflammatory cytokines.	TCR engineered T-cells HBV+ HepG2 cells	T-cells are dependent on tumor cells for migration and induction of apoptosis. Level of oxygen and cytokines important factor in their optimal activity.	[106]
Multiplexed microfluidic device laden with tumor tissue. Infusion of tumor infiltrating lymphocytes.	MC38 tumors and cells PD38+ T-cells Human tumor tissue CD45+ tumor infiltrating lymphocytes	Presence of anti-PD-1 inhibitor led to higher cell death and infiltration into the tumor tissue.	[107]
Breast cancer cells seeded into type I collagen. Separate microchannels mimicking the lymphatic and blood vessels.	MCF-7 breast cancer cells Microvascular endothelial cells	Research on cutoff pore size, ECM structure and lymphatic drainage showed that extravasation and interstitial diffusion was significantly decreased with particles of 100 to 200 nm (smaller than EPR window).	[108]
Two culture chambers (melanoma and splenocytes compartment) connected via narrow capillary migration channels.	B16.F10 murine melanoma cells Murine splenocytes	Absence of IFN regulatory factor 8 (IRF-8) led to poor splenocyte migration towards and interaction with cancer cells.	[109,110]

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
