# Peer review of "Immune Organs and Immune Cells on a Chip: An Overview of Biomedical Applications"

_micromachines, 2020, doi:10.3390/mi11090849_

Round 1

Reviewer 1 Report

This study summarizes recent advances in the organ chips for the recapitulation of immune systems. The authors started their description with the historical background, proceeded to the physiology. Later on, the authors summarized the examples of immune-on-a-chip with detailed classification. Overall, this review article is well-organized and well-written, and well-matched with the wide readership of the Micromachines journal. I suggest acceptance without further revision.

Author Response

We thank the reviewer for his suggestion to accept without further revision.

Reviewer 2 Report

The authors covered organ-on-chip devices related to immunology study. The review provided an overview of history and complexity of immunology which emphasizes the necessity of comprehensive in vitro models, such as OoC. It is well categorized to cover most topics that have been discussed in the field. However, it would be a great benefit to address following points for readers as well as the OoC research community.  

  1. Throughout the review, the authors strongly emphasize the importance of recapitulating complexity and 3D nature of in vivo immune responses. However, it is equally important to simplify the pathophysiology to make the device readily usable and manufacturable. In addition, some pathophysiology doesn’t have to be mimicked in 3D context for sake of simplicity. Indeed, many OoC references that the authors introduced in the paper include at least partly 2D, and for many biological questions, simple transwell assay is still the go-to method. Thus, it is important to discuss in detail about what biology we want to mimic in vitro and why it is significant and what microenvironmental components are important to mimic the specific pathophysiology. For example, early studies describing neutrophil chemotaxis in microfluidic gradient chamber [Jeon et al., Nature Biotechnology, 2002] addressed the question that had been known but impossible to prove in a precisely controlled way. The authors could describe these points in detail with examples in separate sections.

  1. It would be beneficial to put some milestone microfluidic researches to address immunological questions together with the general history of immunology in the introduction. The authors also could describe what early efforts have been devoted to recapitulating basic immunological components on chip. (e.g. blood and lymphatic vessels, 3D environment of immune cells). 

  1. In future perspective, the author should mention recent efforts in the field to translate PDMS-based academic research to commercialized products, including injection-molding based devices, though those devices have not been used widely in immunology studies yet.  

Author Response

We thank the reviewer for their comments and valuable input for this manuscript. Detailed description of the implemented changes can be found in the file attached.
